

# Assessment of functional fitness impacted by hospital rehabilitation in post-stroke patients who additionally contracted COVID-19

Justyna Leszczak, Joanna Pyzińska, Joanna Baran, Rafał Baran, Krzysztof Bylicki and Teresa Pop

Institute of Health Sciences, Medical Faculty, University of Rzeszów, Rzeszów, Poland

## ABSTRACT

**Background:** The aim of the study was to assess the effects of rehabilitation in post-stroke patients, or post-stroke patients with simultaneous COVID-19 infection, in relation to: improved locomotion efficiency, improved balance, reduced risk of falling as well as the patients' more effective performance in everyday activities.

**Methods:** The study involved 60 patients in the early period (2–3 months) after a stroke. Group I consisted of 18 patients (30.0%) who, in addition to a stroke, also contracted COVID-19. Group II consisted of 42 patients (70%) post-stroke, with no SARS-CoV2 infection. The effects were assessed on the basis of: Tinetti test, Timed Up & Go test and Barthel scale.

**Results:** Both groups achieved a statistically significant improvement in their Barthel score after therapy ($p < 0.001$). The Tinetti test, assessing gait and balance, showed that participants in Group I improved their score by an average of 4.22 points. ±4.35, and in Group II, on average, by 3.48 points ± 3.45 points. In the Timed Up & Go test over a distance of 3 m, significant improvement was achieved in both groups, as well but the effect was higher in Group I ($p < 0.001$).

**Conclusions:** Hospital rehabilitation in the early period after stroke improved locomotion efficiency and balance, and reduced the risk of falls in post-stroke patients, both with and without COVID-19 infection.

Corresponding author
Justyna Leszczak,
jleszczak@ur.edu.pl

## INTRODUCTION

Stroke, according to the World Health Organization (WHO), is a sudden onset of focal or global brain dysfunction, lasting more than 24 h (if it does not lead to death earlier) attributed exclusively to causes of vascular origin, affecting the cerebral blood flow (*Aho et al., 1980*). Approximately 15 million people worldwide suffer a stroke every year. Out of those, 5 million people die, whereas 5 million survivors are permanently disabled and need assistance from other people (*World Health Organization (WHO), 2023*). It is also estimated than in Poland approximately 90,000 people experience ischemic stroke each year (*Filipska et al., 2019*).

Stroke is the second most common cause of both disability and death in the world, with the highest burden of disease falling on low- and middle-income countries. In 2016, there were 13.7 million new strokes worldwide (*Saini, Guada & Yavagal, 2021*).

Despite the decline in stroke mortality observed in recent years, the global burden of stroke is increasing. Therefore, a more comprehensive approach to primary stroke prevention is needed, along with extensive rehabilitation applied as early as possible (*Coleman et al., 2017*; *Irisawa & Mizushima, 2020*).

The problems experienced post-stroke are mainly associated with complications which lead to partial or total deterioration of the patient's fitness. Functional fitness is related to the ability to perform activities of daily living, such as walking, climbing stairs, dressing, eating, washing, preparing meals, *etc*. Due to the fact that individuals become disabled after a stroke, their functional performance is impaired (*Saunders et al., 2020*; *Pinedo et al., 2014*). Following stroke, physical capacities of the survivors are affected as a consequence of various complications such as paresis, paralysis, balance and gait disorders, risk of falls, reduced functional fitness, difficulty swallowing, sensory, vision and hearing disorders, difficulty in recognising one's own body, impaired speech, difficulty in reading and writing (*Lo Coco, Lopez & Corrao, 2016*; *Pula & Yuen, 2017*; *Ballesteros Pomar & Palazuelo Amez, 2017*; *Batchelor et al., 2012*; *Plantin et al., 2021*). Impaired balance and risk of falling are among the main problems experienced by stroke survivors (*Pérez-de la Cruz, 2021*). According to researchers, the risk of falling may be linked to cognitive deficiencies rather than impaired motor patterns (*Hollands et al., 2010*). In the related literature one can find recommendations related to preventing falls in individuals with neurological disorders. It has been reported that the turning duration in the Timed Up & Go Test (TUG) and the scores in the Mini Balance Evaluation Systems Test (Mini-BESTest) predict the chance of being a faller (*Caronni et al., 2023*). This indeed is a very important practical information from the viewpoint of preventing falls.

Similarly, the COVID-19 pandemic, which began in 2019, threatened human life and health (*Ejaz et al., 2020*). Infection with SARS-CoV-2 first manifested with mild or severe symptoms similar to pneumonia (*Khan et al., 2020*). Subsequent variants of the disease presented a number of different symptoms such as sore throat, weakness, loss of taste and smell. Some patients had an asymptomatic infection. In particular, people with comorbidities, such as diabetes (*Rao, Lau & So, 2020*; *Fernandez et al., 2018*), obesity (*Zhang et al., 2018*; *Ryan, Ravussin & Heymsfield, 2020*; *World Health Organization (WHO), 2020*), chronic obstructive pulmonary disease (COPD) (*Qiu et al., 2020*; *Wan et al., 2020*; *Liu et al., 2020a*), cardiovascular diseases (CVD) (*Chan et al., 2003*; *Badawi & Ryoo, 2016*; *Zhou et al., 2020*; *Zheng et al., 2020*), stroke, hypertension (*Chen et al., 2020a*; *Fang, Karakiulakis & Roth, 2020*), malignant tumours (*Chen et al., 2020b*; *Wang et al., 2020*) and other comorbidities, are more vulnerable to the COVID-19 infection. As a result of its additional burden on the body, COVID-19 may present a threat to life or lead to disability, reduced quality of life and decrease in functional fitness in those who survive the disease. It was also observed that during the COVID-19 pandemic older people more
frequently experienced low-energy injuries (due to falls or slips) (*Gawronska & Lorkowski, 2021*). Similar findings were reported by researchers who investigated the effect of COVID-19 on the incidence of fractures in Germany, and observed a higher incidence of fractures after the outbreak of the pandemic (*Heinz et al., 2023*). A study has also shown that patients with stroke combined with COVID-19 present poorer functional performance compared to non-COVID-19 patients with stroke (*Ntaios et al., 2020*). After the outbreak of the COVID-19 pandemic, thrombotic complications and ischemic stroke were observed and documented in COVID-19 patients (*Sagris et al., 2021*). However, it was also observed that the overall number of stroke-related admissions decreased during the pandemic, possibly because patients experiencing stroke symptoms, particularly if these were mild, did not consult a doctor, due to a fear of contracting the coronavirus (*Oxley et al., 2020*; *Bangalore et al., 2020*). Stroke is one of the leading causes of disability and the second most common cause of death, so it is necessary to promptly implement early intervention to prevent these outcomes, through rehabilitation training programs (*Avan et al., 2019*; *Norlander et al., 2018*; *Donkor, 2018*; *Cieza et al., 2020*). During the COVID-19 pandemic, one of the problems regarding the rehabilitation of stroke patients was linked with the lack of rehabilitation professionals; there were also restrictions imposed by governments and resulting in less frequent in-person contact with COVID-19 patients. Because of this, during the COVID-19 pandemic the Augmented Reality Rehabilitation System (AR Rehab) was introduced and proved to be effective, at the same time reducing the trainers' effort and decreasing the contact rate (*Yang et al., 2022*). Individuals affected by stroke experience cognitive impairment, which adversely impacts their social functioning and the quality of life (*El Husseini et al., 2023*). Other factors adversely affecting the patients' quality of life include social isolation and reduced direct contacts during the COVID-19 pandemic (*Huang et al., 2022*). Notwithstanding the published reports related to continued rehabilitation with reduced in-person contact, and its positive aspects, the very fact that direct contacts were reduced led to decrease in the quality of life in patients with COVID-19 (*Dragioti et al., 2022*; *Crosetti et al., 2021*; *Yang et al., 2022*). Many research reports available in the literature discuss general health-related effects of COVID-19. It has been noted, however, that there are few studies investigating the effect of COVID-19 on functional capacities in post-stroke patients. Similarly, numerous research projects have assessed progress in rehabilitation of patients after a stroke (*Liu, Yin & Cai, 2022*), whereas few studies have focused on assessing patients with stroke and COVID-19 (*Herrera-Hernández et al., 2023*). Many studies report higher mortality rates and discuss a need for meticulous medical care in this group of patients, whereas there is a lack of information about in-person rehabilitation applied and its effectiveness. The aim of the study was to assess the effects of rehabilitation in post-stroke patients or post-stroke patients with simultaneous COVID-19 infection in relation to: improving locomotion efficiency, improving balance, reducing the risk of falling and helping the patient perform everyday activities.

## MATERIALS AND METHODS

### Ethics

The study was conducted as a prospective observational study.

The study was conducted in accordance with the ethical rules specified in Declaration of Helsinki, and the protocol was approved by the Local Bioethics Commission of the University of Rzeszow (consent no. 2020/10/22). Written consent was obtained from all participants in the study. The study was registered in the Open Science Framework (OSF) Registration DOI 10.17605/OSF.IO/NPUVY (*Open Science Framework (OSF), 2021*).

### Study design and participants

Prior to the study, the sample size was calculated based on the annual number of admissions of stroke patients to the Department of Neurology with Stroke Department at the Hospital in the Podkarpackie Region in Poland. The examinations were carried out between October 2021 and February 2022. The sample size was calculated for a 95% confidence interval, a significance level of 0.05 and a maximum error of 10%. Based on the inclusion and exclusion criteria, the study group enrolled for the study comprised 60 patients in the early period (2–3 months) after ischemic stroke, admitted to the neurology department of the regional hospital.

Inclusion criteria: ischemic stroke (internal capsule stroke, middle cerebral artery), first complete stroke, ability to stand without help, onimpairment of higher mental functions (MMSE scale 27 points), completion of a 3-week rehabilitation program, informed consent of the patient to participate in the study.

Exclusion criteria: incomplete stroke (*e.g.*, transient ischemic attack, TIA), haemorrhagic stroke, second or subsequent stroke, inability to stand independently (balance disorders and dizziness), ischemic changes located in the cerebellum and brain stem, lower limb injuries after stroke, diagnosis of other conditions, such as unstable medical condition, orthopaedic or rheumatic diseases, oncological diseases and other neurological diseases (*e.g.*, multiple sclerosis), pain as well as musculoskeletal inflammation (Fig. 1).

The study involved 60 post-stroke patients. The participants were divided into two groups. The first group consisted of 18 patients (30.0%) who, in addition to a stroke, also contracted COVID-19. The second group (70%) consisted of post-stroke patients withoutSARS CoV-2 infection.

The patients with stroke and COVID-19 on average were younger than the patients in the second group ($p = 0.031$). The mean age of the patients with stroke and COVID-19 was 61.56 years, and those with stroke and no COVID-19 on average were aged 69.88 years. The BMI values of identified in the two groups werenot significantly different and ranged from 17.15 to 40.39 kg/m$^2$ (Table 1).

The whole group included 31 women and 29 men. The participants from the two subgroups did not differ significantly in terms of education ($p = 0.0856$), marital status

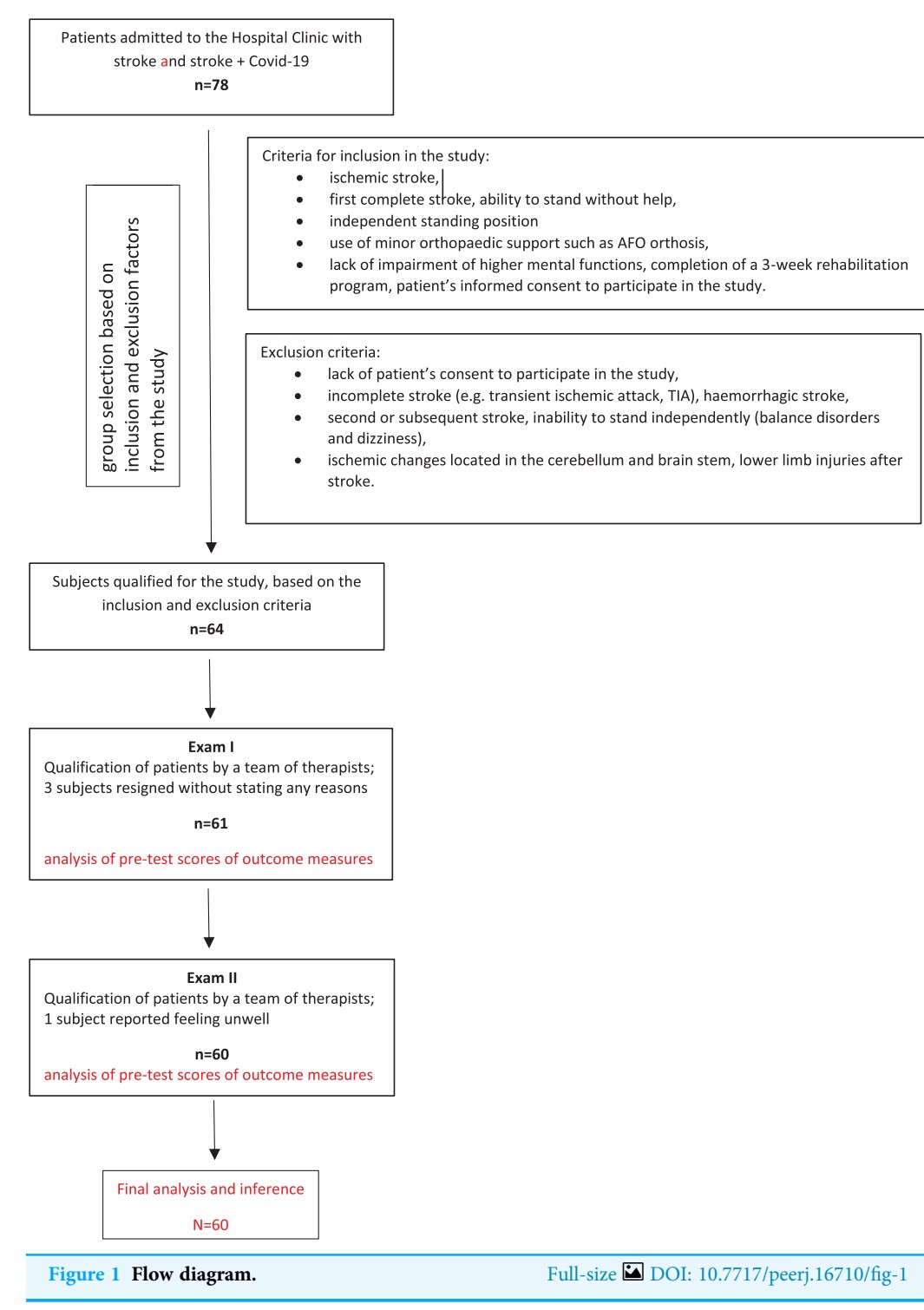

**Figure 1  Flow diagram.**

($p = 0.950$), occupational status ($p = 0.538$) or existing chronic diseases ($p = 0.431$). The co-existing conditions diagnosed in the participants included hypertension, diabetes, atrial fibrillation, Parkinson's disease, obesity, atherosclerosis and circulatory insufficiency. See Table 2 for details.

**Table 1 Characteristics of the study group.**

| | Basic descriptive statistics | | | $p$ |
|---|---|---|---|---|
| | N | $\bar{x}$ | SD | |
| **Age** | | | | |
| Covid + Stroke | 18 | 61.56 | 17.53 | t = −2.21 $p$ = 0.031 |
| Stroke | 42 | 69.88 | 11.16 | |
| Total | 60 | 67.38 | 13.78 | |
| **BMI** | | | | |
| Covid + Stroke | 18 | 25.46 | 5.13 | t = −0.74 $p$ = 0.462 |
| Stroke | 42 | 26.67 | 6.03 | |
| Total | 60 | 26.31 | 5.76 | |

Note:
t, the Student's t-test value for independent variables; $p$, test probability indicator.

**Table 2 Socio-demographic characteristics of the participants.**

| | Covid + Stroke | | Stroke | | Total | | $p$ |
|---|---|---|---|---|---|---|---|
| | N | % | N | % | N | % | |
| **Sex** | | | | | | | |
| Women | 11 | 61.1% | 20 | 47.6% | 31 | 51.7% | $\chi^2(1)$ = 0.92 $p$ = 0.337 |
| Men | 7 | 38.9% | 22 | 52.4% | 29 | 48.3% | |
| Total | 18 | 100.0% | 42 | 100.0% | 60 | 100.0% | |
| **Education** | | | | | | | |
| Basic | 3 | 16.7% | 5 | 11.9% | 8 | 13.3% | $\chi^2(3)$ = 0.77 $p$ = 0.856 |
| Essential vocational | 5 | 27.8% | 9 | 21.4% | 14 | 23.3% | |
| Medium | 8 | 44.4% | 21 | 50.0% | 29 | 48.3% | |
| Higher | 2 | 11.1% | 7 | 16.7% | 9 | 15.0% | |
| Total | 18 | 100.0% | 42 | 100.0% | 60 | 100.0% | |
| **Marital status** | | | | | | | |
| Free | 5 | 27.8% | 12 | 28.6% | 17 | 28.3% | $\chi^2(1)$ = 0.00 $p$ = 0.950 |
| In relation with | 13 | 72.2% | 30 | 71.4% | 43 | 71.7% | |
| Total | 18 | 100.0% | 42 | 100.0% | 60 | 100.0% | |
| **Professional activity** | | | | | | | |
| Professionally active | 3 | 16.7% | 10 | 23.8% | 13 | 21.7% | $\chi^2(1)$ = 0.38 $p$ = 0.538 |
| Professionally inactive | 15 | 83.3% | 32 | 76.2% | 47 | 78.3% | |
| Total | 18 | 100.0% | 42 | 100.0% | 60 | 100.0% | |
| **Chronic diseases** | | | | | | | |
| Yes | 11 | 61.1% | 30 | 71.4% | 41 | 68.3% | $\chi^2(1)$ = 0.62 $p$ = 0.431 |
| No | 7 | 38.9% | 12 | 28.6% | 19 | 31.7% | |
| Total | 18 | 100.0% | 42 | 100.0% | 60 | 100.0% | |

| Table 2 (continued) | Covid + Stroke | | Stroke | | Total | | p |
|---|---|---|---|---|---|---|---|
| | N | % | N | % | N | % | |
| **Chronic diseases*** | | | | | | | |
| Hypertension | 6 | 54.5% | 19 | 63.3% | 25 | 60.1% | – |
| Diabetes | 4 | 36.4% | 10 | 33.3% | 14 | 34.1% | |
| Atrial fibrillation | 0 | 0.0% | 1 | 3.3% | 1 | 2.4% | |
| Parkinson's disease | 0 | 0.0% | 1 | 3.3% | 1 | 2.4% | |
| Obesity | 5 | 45.5% | 12 | 40.0% | 17 | 41.5% | |
| Atherosclerosis | 3 | 27.3% | 8 | 26.7% | 11 | 26.8% | |
| Circulation insufficiency | 2 | 18.2% | 7 | 23.3% | 9 | 21.9% | |

**Notes:**
$\chi^2$, Pearson chi-square test value; p, test probability indicator.
* The percentage of comorbidities was presented as a percentage of people reporting the occurrence of comorbidities in each group. Some people reported the occurrence of several comorbidities at the same time, therefore the numbers and percentages indicate the number of responses recorded, not the number of patients.

## Procedure

All the participants were subjected to examinations twice: before the rehabilitation process and at the end of a three-week comprehensive rehabilitation program. Three outcome measures were applied in the study.

The Timed Up & Go gait test, introduced by *Podsiadło & Richardson (1991)*, is designed to assess functional capacities and risk of falling. At the command "START", the subject is to: 1. get up from the chair, 2. walk the distance of 3 m at a normal pace, 3. cross the line ending the designated distance, 4. make a turn of 180 degrees, 5. return to the chair and assume a sitting position again. RESULTS: below 10 s–correct normal, functional efficiency (low risk of falling); 10–19 s-the subject can go outside on his/her own, and does not need any walking aids, he/she is independent in most activities of daily living, however, an in-depth assessment of the risk of falling is recommended (moderate risk of falling); 19 s or more-significantly limited functional capacities, the subject cannot go outside alone, auxiliary equipment for walking is recommended, (high risk of falling) (*Son & Park, 2019*, *Besios et al., 2019*; *Podsiadło & Richardson, 1991*).

The Tinetti Mobility Test (TMT) is used to assess balance and gait. The patient can score up to 16 points in the balance assessment, and 12 points in the gait assessment, which gives a total of 28 points. A score of less than 26 indicates that there is a problem. On the other hand, a score of less than 19 points means a 5 times higher risk of falling than a score of 28 points. Moreover, a score of "0" points in at least one item, or a score of "1" point in at least two items (if it is not the highest number of points to be scored) is an indication for a consultation with a physiotherapist (*Köpke & Meyer, 2006*; *Browne & Nair, 2019*, *Besios et al., 2019*; *Podsiadło & Richardson, 1991*).

The Barthel scale enables assessment of the patients' performance in the basic activities of daily living while they are learning to walk in the early stages after stroke. Based on the

patients' scores, their condition is described in the following way: 0–20 points, understood as patient's severe dependency, 21–80 points–partial lack of independence, and 81 or more points reflecting patient's independence. A maximum of 100 points can be scored on Barthel scale (*Musa & Keegan, 2018*; *Liu et al., 2020b*).

*Besios et al. (2019)* assessed validity of TMT and TUG in patients with neurological disorders and reported that both tools were reliable in assessing mobility and functional fitness in patients with hemiparesis and with multiple sclerosis. Importantly, in their study the researchers noticed that some patients performing TUG test found it difficult to return and sit back on the chair because of considerable spasticity and impaired balance (*Besios et al., 2019*). Although TUG is generally known to be a reliable outcome measure in individuals with neurological impairments, *Abdollahi et al. (2023)* performed detailed assessment of movements performed during a TUG trial; for this purpose they applied a motor-cognitive dual-task model, and they used eight motion sensors attached to the subjects' feet, shanks, legs, sacrum and sternum. By using this procedure they were able to identify detailed gait parameters in the 'walking portions', *i.e.*, the specific sections of a TUG trial; these findings may be helpful in designing effective screening techniques to be applied in patients with neurological problems (*Abdollahi et al., 2023*).

Socio-demographic and socio-economic characteristics (such as age, sex, occupational status, marital status) were self-reported by the participants in the first part of the questionnaire.

## Rehabilitation program

The rehabilitation program were continued from Monday to Friday for 3 weeks. During the rehabilitation stay, patients took part in individual exercises, including morning and breathing exercises. In order to improve impaired motor functions, patients had the PNF method and NDT Bobath neurodevelopmental therapy for adults implemented.

## Statistical analysis

The obtained material was subjected to standard statistical analysis procedures in the Statistica 13.1 package (TIBCO Software, Palo Alto, CA, USA), assuming a level of statistical significance of $p < 0.05$.

First, the normality of distribution was assessed using the Shapiro Wilk test. In the case of variables meeting the assumptions of normal distribution, parametric tests were used: Student's t-test for independent samples (age, BMI, Barthel scale results), Student's t-test for dependent variables (Timed Up & Go test results) and in the absence of normality of distribution-non-parametric tests: the Mann Whitney U test (results of the Timed Up & Go test) comparing two populations or the non-parametric Wilcoxon paired rank test (assessment of independence based on the Barthel scale, results of the Tinetti test, results of the Timed Up & Go test) to assess intra-group variability in both populations. Qualitative data, such as characteristics of the groups in terms of sociodemographic factors, *e.g.*, gender, education, marital and professional status, existing chronic diseases, were developed using the Pearson chi-square test.

**Table 3 Results on the Barthel scale before and after therapy in two groups.**

| Barthel | COVID + Stroke | | Stroke | | $p$ | |
|---|---|---|---|---|---|---|
| | $\bar{x}$ | SD | $\bar{x}$ | SD | $t_1$ | $p$ |
| Before therapy | 47.2 | 21.5 | 42.4 | 21.1 | 0.81 | 0.421 |
| After therapy | 66.4 | 28.2 | 56.6 | 23.0 | 1.40 | 0.166 |
| Effect | 19.2 | 16.9 | 14.3 | 9.5 | 1.43 | 0.156 |
| $p$ | $t_2 = -4.80$ $p < 0.001$ | | $t_2 = -9.77$ $p < 0.001$ | | | |

Note:
$t_2$, Student's t-test value for dependent variables; $p$, test probability indicator.

**Table 4 Assessment of independence based on the Barthel scale.**

| Assessment of independence | COVID + Stroke | | | | Stroke | | | |
|---|---|---|---|---|---|---|---|---|
| | Before therapy | | After therapy | | Before therapy | | After therapy | |
| | N | % | N | % | N | % | N | % |
| Completely dependent (0–20) | 3 | 16.7% | 2 | 11.1% | 9 | 21.4% | 2 | 4.8% |
| Partial lack of independence (21–80) | 15 | 83.3% | 11 | 61.1% | 33 | 78.6% | 37 | 88.1% |
| Independent (81–100) | 0 | 0.0% | 5 | 27.8% | 0 | 0.0% | 3 | 7.1% |
| Total | 18 | 100.0% | 18 | 100.0% | 42 | 100.0% | 42 | 100.0% |
| $p$ | $Z_2 = 1.89$ $p = 0.058$ | | | | $Z_2 = 2.80$ $p = 0.005$ | | | |

Note:
$Z_2$, value of Wilcoxon pair rank test; $p$, test probability indicator

# RESULTS

Both stroke/COVID-19 patients and stroke patients achieved a statistically significant improvement in Barthel scores after therapy ($p < 0.001$). Post-stroke patients with simultaneous COVID-19 infection achieved a greater effect than those after stroke alone (19.2 *vs* 14.3) (Table 3).

The scores acquired on the Barthel scale before the therapy, by both stroke/COVID-19 patients and stroke patients showed that most study participants required help from others in performing activities of daily living (83.3% and 78.6% respectively). After the therapy program the change in the level of independence in the stroke/COVID-19 group was close to the threshold of statistical significance ($p = 0.058$) and in the post-stroke group it was statistically significant ($p = 0.005$) (Table 4).

The statistical analysis showed that rehabilitation effects in gait and balance achieved by patients with stroke and COVID-19 were more than twice as good as the effects observed in those without COVID-19 infection. This was the case in both the overall assessment and in the specific tests (gait and balance). In all these measurement, statistically significant differences were observed between the results before and after therapy in both stroke/COVID-19 group and stroke group. In the gait test the differences between the results before and after the therapy were at a significance level of $p = 0.004$ in the group of patients with stroke and COVID-19 and $p < 0.001$ in the group of stroke patients without COVID-19. In the balance test the differences in both groups were at a significance level of $p < 0.001$. In the combined assessment of gait and balance the differences were at a

**Table 5 The Tinetti test.**

| | COVID + Stroke | | | | | Stroke | | | | | p | |
|---|---|---|---|---|---|---|---|---|---|---|---|---|
| | $\bar{x}$ | Me | Q1 | Q3 | SD | $\bar{x}$ | Me | Q1 | Q3 | SD | $Z_1$ | p |
| **Tinetti–gait (0–10 pkt.)** | | | | | | | | | | | | |
| Before therapy | 3.67 | 3.50 | 0.00 | 10.0 | 3.01 | 2.45 | 1.00 | 0.00 | 10.0 | 2.92 | 1.62 | 0.106 |
| After therapy | 5.89 | 6.50 | 0.00 | 10.0 | 3.31 | 4.36 | 4.00 | 0.00 | 10.0 | 3.63 | 1.37 | 0.170 |
| Effect | 2.22 | 2.50 | −3.00 | 5.00 | 2.29 | 1.90 | 1.00 | 0.00 | 7.00 | 1.87 | 0.76 | 0.445 |
| p | $Z_2 = 2.88\ p = 0.004$ | | | | | $Z_2 = 4.93\ p < 0.001$ | | | | | | |
| **Tinetti–balance (0–15 pkt.)** | | | | | | | | | | | | |
| Before therapy | 5.28 | 6.00 | 0.00 | 11.00 | 3.64 | 5.55 | 5.50 | 0.00 | 14.00 | 4.32 | −0.35 | 0.727 |
| After therapy | 9.78 | 11.00 | 1.00 | 14.00 | 3.92 | 7.95 | 9.00 | 0.00 | 15.00 | 4.54 | 1.33 | 0.183 |
| Effect | 4.50 | 4.50 | 0.00 | 9.00 | 2.43 | 2.40 | 2.00 | 0.00 | 7.00 | 1.82 | 3.08 | 0.002 |
| p | $Z_2 = 3.62\ p < 0.001$ | | | | | $Z_2 = 5.01\ p < 0.001$ | | | | | | |
| **Tinetti—total (0–35 pkt.)** | | | | | | | | | | | | |
| Before therapy | 7.33 | 7.50 | 0.00 | 19.00 | 5.92 | 4.74 | 2.00 | 0.00 | 19.00 | 5.60 | 1.72 | 0.086 |
| After therapy | 11.56 | 12.50 | 0.00 | 22.00 | 6.60 | 8.21 | 7.50 | 0.00 | 19.00 | 6.85 | 1.49 | 0.135 |
| Effect | 4.22 | 4.50 | −5.00 | 10.00 | 4.35 | 3.48 | 2.00 | 0.00 | 13.00 | 3.45 | 0.79 | 0.428 |
| | $Z_2 = 2.95\ p = 0.003$ | | | | | $Z_2 = 4.93\ p < 0.001$ | | | | | | |

Note:
$Z_2$, Wilcoxon pairwise order test value; p, test probability index.

**Table 6 Timed up & go test over a distance of 3 m.**

| | COVID + Stroke | | | | | Stroke | | | | | p | |
|---|---|---|---|---|---|---|---|---|---|---|---|---|
| | $\bar{x}$ | Me | Q1 | Q3 | SD | $\bar{x}$ | Me | Q1 | Q3 | SD | U | p |
| **Time up & go** | | | | | | | | | | | | |
| Before therapy | 13.56 | 13.50 | 12.50 | 14.50 | 1.07 | 12.98 | 12.50 | 11.50 | 15.00 | 1.81 | 297.50 | 0.197 |
| After therapy | 8.64 | 8.75 | 8.50 | 9.00 | 0.41 | 10.07 | 10.00 | 9.00 | 11.00 | 1.30 | 132.50 | <0.001 |
| Effect | −4.92 | −5.00 | −6.00 | −4.00 | 1.22 | −2.90 | −3.00 | −4.00 | −1.00 | 1.97 | 159.00 | <0.001 |
| p | $t = 17.16\ p < 0.001$ | | | | | $Z = 5.26\ p < 0.001$ | | | | | | |

Note:
U, Mann Whitney U-test; T, student t-test for dependent samples; Z, Wilcoxon test.

significance level of $p = 0.003$ in the group of patients with stroke and COVID-19 and $p < 0.001$ in the group of stroke patients without COVID-19. The detailed results are shown in Table 5.

In the Timed Up & Go test over a distance of 3 m, patients from the two groups did not differ significantly in the measurement before therapy, whereas the second measurement showed better values in this test in the stroke/COVID-19 group. Therefore, the effect of the treatment was greater in the latter subgroup, which was confirmed by the results that were statistically significant, at a level of the $p < 0.001$. Differences in the improvement achieved as a result of the treatment were found in the participants from both groups ($p < 0.05$). More specifically, the scores improved by an average of 4.92 s in stroke/COVID-19 patients, and stroke patients by an average of 2.9 s in stroke patients. The results in both groups were at a level of $p < 0.001$ level (Table 6).

## DISCUSSION

The aim of the study was to assess the effects of rehabilitation in patients with stroke, as well as patients with stroke and COVID-19 infection, in relation to: locomotion efficiency, balance, the risk of falling and performance in activities of daily living. Studies show that stroke is the most common cause of disability in the adult population. Notably, the incidence of this condition is increasing in young people, which has enormous social and economic consequences (*Smajlović, 2015*; *Guzik et al., 2019*). The most important element in comprehensive and long-term care post-stroke include early rehabilitation, as well as programs making it possible for the patients to recover independence in the activities of daily living. To achieve patients commonly should participate in long-lasting and continued training to regain self-care ability. *Belagaje (2017)* as well as *Shaikh et al. (2022)* in their studies, found that comprehensive, early and continuous rehabilitation facilitates functional improvement in post-stroke patients.

There are many studies assessing functional fitness in patients at an early stage after a stroke, reflecting effects of ongoing rehabilitation. Following rehabilitation programs, researchers have observed a reduction in neurological deficit, improvement in walking and self-care skills, and reduction in the degree of disability (*Duncan, Lai & Keighley, 2000*; *Lee et al., 2015*; *Dąbrowski et al., 2019*; *Kopp et al., 1999*). Studies have also shown that Barthel Index, Activities of Daily Living (ADL) scores, as well as Generic Quality of Life Inventory-74 (GQOLI-74) scores in post-stroke patients during the early post-hospital rehabilitation period were higher compared to the pre-intervention scores (in all cases $p < 0.05$) (*Yu et al., 2021*). Similarly, *Rahayu et al. (2020)* reported improvements in functional capacities and balance in stroke patients following a seven-day physiotherapy intervention.

In the available literature there are reports discussing the potential of rehabilitation and the characteristics of stroke incidents in patients with COVID-19. The authors report that higher mortality rates, poorer ability to recover neurological functions as well as severe disability are observed in patients with stroke and COVID-19 (*Pomnikov et al., 2021*). However, there are still very few studies assessing the progress achieved through rehabilitation by post-stroke patients, taking into account the factor of additional COVID-19 illness. In the present studie, the functional fitness was assessed usingthe Barthel scale. The findings show improved performance in both stroke/COVID-19 patients and stroke patients after rehabilitation program administered in hospital. *Asirvatham et al. (2022)* examined post-stroke patients with COVID-19, showed that there was an increase in functional gain (mean FIM gain, $32.9 \pm 8.9$) and an improvement in functional capacities during active rehabilitation (LOS, $62.45 \pm 37.61$). The authors suggest that active rehabilitation and immediate intervention be required to rehabilitate post-stroke patients after COVID-19 infection, who are a sensitive population, in order to enable adequate functional improvement (*Asirvatham et al., 2022*). However, *Pomnikov et al. (2021)* in their study showed higher mortality rates in the population of stroke patients who were infected with COVID-19; they also reported, decreased ability of such patients to recover of neurological functions and a greater risk of significant disability. In a global report on stroke and COVID-19, *Ntaios et al. (2020)* investigated outcomes of patients with

laboratory-confirmed COVID-19 and acute ischemic stroke hospitalized 28 facilities in 16 countries. Their findings showed that, compared to non-COVID-19 stroke patients, the COVID-19/stroke patients had a higher risk of severe disability (median mRS 4 [IQR, 2-6] compared to median mRS 2 [IQR, 1-4], $p < 0.001$) and death (odds ratio, 4.3 95% CI [2.22–8.30]) (*Ntaios et al., 2020*).

In their study, *Chou et al. (2021)* examined functional capacities and balance in post-stroke patients in two non-pandemic groups (82 individuals) and a pandemic groups (11 individuals), and they observed that following a PAC program, which was aimed at improving motor functions, performance in activities of daily living, quality of life, oral function and mental health, the non-pandemic group showed significant improvement in all the measures of functional status, including: Barthel Index (BI) effect size ES-2: 0.63, EuroQoL-5 Dimension (EQ-5D) ES-2:-0.55, Lawton–Brody instrumental activities of daily living (IADL) ES-2: 0.61, Berg Balance Scale (BBS) ES-2: 0.63, 5 m walking speed (5MWS) ES-2: 0.54 and 6-min walking distance (6MWD) ES-2: 0.54. The pandemic group showed significant improvement in most functional status indicators after the PAC intervention, including: BI (ES: 0.57), IADL (ES: 0.52):, BBS (ES: 0.63), 5MWS (ES: 0.57) and 6MWD (ES: 0.57). The results of EQ-5D did not differ significantly in the pandemic group ($p = 0.13$) (*Chou et al., 2021*). Similar improvements in functional fitness measured using the Barthel scale, balance and gait assessed with the Tinetti test, and the risk of falling measured using the Timed Up & Go test were observed in the present studies.

Because of the regime introduced by the local government and the hospitals at the time of the research project, the tests applied in the study were of a subjective nature. Indeed, other studies available in the literature apply sensors and equipment enabling objective measurement of performance using instrumented tests (*Bonnyaud et al., 2015*; *Abdollahi et al., 2022*; *Bower et al., 2019*) and these should be used in further research and in follow-up studies.

In general, the current findings prove that it is important to introduce hospital rehabilitation as early as possible in post-stroke patients, and in post-stroke/COVID-19 patients in order to make it possible for them to recover lost functions as quickly as possible.

## LIMITATIONS

Despite great efforts, the authors of this study did not avoid certain limitations, starting with the small number of study participants. However, the sample size calculation which was carried out showed that our sample of participants is adequately powered for the feasibility study. Secondly, the material for the analyses was acquired using subjective questionnaires/scales and no diagnostic equipment. The fact that the latter tools could not be used for acquiring objective data can be attributed to the restrictions imposed due to the COVID-19 pandemic and the resulting prohibition on any exchange between hospital wards and laboratories. However, all questionnaires and scales used in the study are standardised as a result of which the research could be carried out correctly.

## CONCLUSIONS

Inpatient rehabilitation at an early stage post-stroke improved locomotion efficiency and balance, and reduced the risk of falls in post-stroke patients, both with and without COVID-19 infection. Rehabilitation administered to patients at an early stage after stroke allows for the recovery of lost functions as quickly as possible.

## ABBREVIATIONS

| | |
|---|---|
| **WHO** | World Health Organization |
| **TUG** | Timed Up & Go Test |
| **Mini-BESTest** | Mini Balance Evaluation Systems Test |
| **COPD** | Chronic Obstructive Pulmonary Disease |
| **CVD** | Cardiovascular diseases |
| **OSF** | Open Science Framework |
| **MMSE** | Mini–Mental State Examination |
| **TIA** | Transient Ischemic Attack |
| **BMI** | Body Mass Index |
| **TMT** | The Tinetti Test |
| **PNF** | Proprioceptive Neuromuscular Facilitation |
| **ADL** | Activities Of Daily Living |
| **GQOLI-74** | Generic Quality Of Life Inventory-74 |

## ACKNOWLEDGEMENTS

The authors are most grateful to all participants for their committed involvement in the study protocol, despite numerous inconveniences.

### Funding

The authors received no funding for this work.

### Competing Interests

The authors declare that they have no competing interests.

### Author Contributions

- Justyna Leszczak analyzed the data, authored or reviewed drafts of the article, and approved the final draft.
- Joanna Pyzińska conceived and designed the experiments, performed the experiments, prepared figures and/or tables, and approved the final draft.
- Joanna Baran performed the experiments, prepared figures and/or tables, and approved the final draft.
- Rafał Baran analyzed the data, prepared figures and/or tables, and approved the final draft.

- Krzysztof Bylicki performed the experiments, authored or reviewed drafts of the article, and approved the final draft.
- Teresa Pop conceived and designed the experiments, authored or reviewed drafts of the article, and approved the final draft.

## Human Ethics

The following information was supplied relating to ethical approvals (*i.e.*, approving body and any reference numbers):

The study was conducted in accordance with the ethical rules of the Helsinki Declaration and approved by the Local Bioethics Commission of the University of Rzeszow (consent no. 2020/10/22).

## Data Availability

The COVID-19 stroke data is available in the Supplemental File.

## Supplemental Information

Supplemental information for this article can be found online at http://dx.doi.org/10.7717/peerj.16710#supplemental-information.

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
