# Peer review of "Assessment of functional fitness impacted by hospital rehabilitation in post-stroke patients who additionally contracted COVID-19"

_PeerJ, doi:10.7717/peerj.16710_

## Round 0.1 · original submission · Major Revisions

I have now received the reviewers' comments on your manuscript. They have suggested some revisions to your manuscript. Therefore, I invite you to respond to the reviewers' comments and revise your manuscript.

·

Basic reporting

• Your abstract needs more detail. I suggest that you be more specific in descriptions as the abstract is one of the most important areas that give an overall picture of your research. Line 11: mention the duration following stroke (the term “early” needs more specification); line 13: mention the number of post-stroke patients and the percentage.

• Abbreviations should be added

Introduction:
• Kindly provide references for burden of stroke in low and middle-income countries in line 28.
• Line 33-34: could be better phrased such as “following stroke, fitness among stroke survivors are affected due to various complications such as paresis, paralysis……. Remove the statement “After stroke, there are often disorders such as” -line 34
• Line 43: rephrase as it is in active tense.
• Line 48-49: rephrase as “are more vulnerable to the COVID-19 infection”
• Introduction: your introduction needs to be more elaborate. You have mentioned about stroke statistics worldwide and the current burden. A mention about various complications post-stroke along with the references is commendable. But the introduction seems incomplete with missing literature reviews. Your introduction would seem more informative and would be understandable to the general audience if you included the following in this format: (1) introduction to stroke and the overall burden. A mention about the burden in your country as compared to worldwide would add more specificity to your research. (2) a brief introduction into post-stroke patients with COVID-19 infection- clinical picture; (3) differentiate between post-stroke and post-stroke with COVID-19 infection- how they vary in terms of their condition, their fitness and quality of life, etc.; (4) a brief description about rehabilitation and its effects in post-stroke patients; (5) a mention about what functional fitness is (along with references for each). This would add more color to the introduction and be more effective in reaching out to international audience reading your research.

Experimental design

Materials and methods:
• Line 72: specify duration after ischemic stroke
• Line 75-76: independent walking – should be quantifiable in terms of a scale for example those walking independently with a functional ambulation category score of 4 or 5.
• Line 76: lack of impairment of higher mental functions: mention scale such as MMSE or MOCA with cut-off scores
• There is no mention about balance criteria; kindly mention the balance status of the patients who will be included in the study.
• There is no mention about the ability to perform everyday activities; kindly mention functional status with regards to performing activities of daily living with scores on a standardized scale such as Barthel index over score 75 to be included should be mentioned.
• Line 79: can remove the statement “lack of patient consent to participate in the study” as its already explained in the inclusion criteria.
• Line 81: since ischemic changes are mentioned; kindly mention in the inclusion criteria specific ischemic stroke- which brain area involvement.
• Add any other medical comorbidities as part of exclusion criteria.
• Your methodology seems more basic and this is another important aspect of the study. To be more precise in your methodology, the following should be mentioned in brief: (1) study setting: where the study was performed; (2) age of the participants ; (3) gender; (4) planned screening or recruitment number; (5) selected number or recruited number; (6) study procedures: how the study populations were approached; how consent was obtained; the language of consent; how once consented, if there was a non-identifiable participant number that was assigned; sociodemographic variables; types of outcome measures; overall duration of the study; and a mention about the expected outcomes of the study.

Validity of the findings

Figure 1:
• change “end” to “and”
• include analysis of pre-test scores of outcome measures after exam 1 and exam 2
• include analysis of post-test scores of outcome measures
• final box can include “analysis and inference”

• line 90-92 should come under table 1 title and not under figure 1

• line 90 : mention stroke patients with COVID-19 infection

• line 91-92: mention p-value for not statistically different. Also, I don’t see a mention about the range 17.5 to 40.39 anywhere in table 1. Kindly clarify.

• Lines 96-100 should come under table 2

• Lines 97-98: mention p values for each of the not statistically significant components.

• Lines 98-100: percentages of co-existing diseases should be mentioned in table 2.

• Line 106: mention where these tests were conducted and by whom.

• Statistical analysis lines 122-133: it would be better if you could mention which statistical tests are used for which variables. In that way, a clearer understanding of the statistical analysis could be obtained.

• Lines 156-158: mention the statistically significant differences of each aspect: Tinetti-gait; Tinetti-balance and Tinetti-total.

• Line 165: remove “this”

• Line 168: change the term treatment to rehabilitation



Discussion

• The discussion is well-written but could be more improvised by adding various other components. You have mentioned about rehabilitation effects and supporting evidences and references. The discussion can be more elaborate with mention about your results about balance and activities of daily living and supporting evidences.

Additional comments

I commend the authors' work in exploring the rehabilitation needs of such vulnerable populations. The entire proposal is well-written, with only a few additions and changes that may be necessary to make the study noteworthy.

Reviewer 2 ·

Basic reporting

1. Line 28: Please provide more recent statistics about the impact of Stroke.
2. Line 39: Several disorders are mentioned in this paragraph. Please specify which of these are specifically known to affect the functional fitness of patients, and increase the risk of falls.
3. Please also provide a further in-depth literature review on the topic of falls in stroke patients since it is related to this study.
4. Line 53: Please mention the previous few studies that reflect the effect of Covid-19 on functional fitness in post-stroke patients. Specifically, what were their methods used, findings from the studies and then add a few statements about how this study offers a novel perspective on the topic.
5. Overall, further details are required on the methods implemented, as well as in introduction and discussion sections to provide the readers with a background on the manner in which such evaluations are conducted.

Experimental design

1. Line 106: Please describe the TUG and Tinetti tests, and the Barthel scale in further detail since not all readers may be familiar with these clinical tests.
2. Please also be more specific on which test/scale assesses which aspect (functional mobility, balance, etc.). Please include the activities and body movements performed during the TUG test. Descriptions and details about the tests can be found in the following references:
a. Besios, T., Nikolaos, A., Vassilios, G., & Giorgos, M. (2019). Comparative Reliability Of Tinetti Mobility Test And Tug Tests In People With Neurological Disorders. International Journal of Physiotherapy, 251-255.
b. Abdollahi, M., Kuber, P. M., Pierce, M., Cristales, K., Dombovy, M., LaLonde, J., & Rashedi, E. (2023, April). Motor-Cognitive Dual-Task Paradigm Affects Timed Up & Go (TUG) Test Outcomes in Stroke Survivors. In 2023 11th International IEEE/EMBS Conference on Neural Engineering (NER) (pp. 1-4). IEEE.

Validity of the findings

1. Line 190: Please include full-forms of the ADL and GQOLI-74 scales here.
2. Line 195: Please mention what was lacking in the previous studies and what efforts are presented in this study to enhance the insights gained.
3. Line 223: Please refer to the outcomes of this study. Mentioning the values here may also be valuable to the readers.
4. Please provide future directions to this study at the end of the discussion section. Currently used tests seem to be subjective. There have been recent efforts to use sensors and equipment to objectively measure the performance using instrumented tests. If the authors agree, this can be one of the future directions. Below are some examples of such studies:
a. Bonnyaud, C., Pradon, D., Vuillerme, N., Bensmail, D., & Roche, N. (2015). Spatiotemporal and kinematic parameters relating to Oriented gait and turn performance in patients with chronic stroke. PLoS ONE, 10(6), 1–14.
b. Abdollahi, M., Kuber, P. M., Shiraishi, M., Soangra, R., & Rashedi, E. (2022). Kinematic Analysis of 360◦ Turning in Stroke Survivors Using Wearable Motion Sensors. Sensors.
c. Bower, K., Thilarajah, S., Pua, Y. H., Williams, G., Tan, D., Mentiplay, B., Denehy, L., & Clark, R. (2019). Dynamic balance and instrumented gait variables are independent predictors of falls following stroke. Journal of NeuroEngineering and Rehabilitation, 16(1), 1–9.
5. There seem to be several co-existing disorders (Line 98), about 60-70%, in the participant pool along with Covid-19. Is there a possibility of an interaction effect of one or more of such disorders affecting the performance of patients more than covid-19? Please comment on this aspect and may possibly be included in the discussion section.

---

## Round 0.2 · accepted · Accept

In my opinion, this manuscript has been revised with attention to the reviewers' comments and can now be published.

Reviewer 2 ·

Basic reporting

Several improvements were made in the manuscript based on prior comments by the reviewers. I appreciate the authors for considering my feedback and I am satisfied with the current version of the manuscript. I have no further comments to add and wish the authors best of luck for their future studies.

Experimental design

N/A

Validity of the findings

N/A

Additional comments

N/A